# Influence of Light of Different Spectral Compositions on Growth Parameters, Photosynthetic Pigment Contents and Gene Expression in Scots Pine Plantlets

**DOI:** 10.3390/ijms24032063

**Published:** 2023-01-20

**Authors:** Pavel Pashkovskiy, Yury Ivanov, Alexandra Ivanova, Vladimir D. Kreslavski, Mikhail Vereshchagin, Polina Tatarkina, Vladimir V. Kuznetsov, Suleyman I. Allakhverdiev

**Affiliations:** 1K.A. Timiryazev Institute of Plant Physiology, Russian Academy of Sciences, Botanicheskaya Street 35, Moscow 127276, Russia; 2Institute of Basic Biological Problems, Russian Academy of Sciences, Institutskaya Street 2, Pushchino 142290, Moscow Region, Russia; 3Department of Plant Physiology, Biological Faculty, Lomonosov Moscow State University, Moscow 119991, Russia

**Keywords:** *Pinus sylvestris* L., seeds, plantlets, light of different spectral composition, photosynthetic pigments, light and hormonal signaling genes

## Abstract

The photoreceptors of red light (phytochromes) and blue light (cryptochromes) impact plant growth and metabolism. However, their action has been barely studied, especially in coniferous plants. Therefore, the influence of blue (maximum 450 nm), red (maximum 660 nm), white light (maxima 450 nm + 575 nm), far-red light (maximum 730 nm), white fluorescent light and dark on seed germination, growth, chlorophyll and carotenoid contents, as well as the transcript levels of genes involved in reception, photosynthesis, light and hormonal signaling of Scots pine plantlets, was investigated. The highest values of dry weight, root length and photosynthetic pigment contents were characteristic of 9-day-old plantlets grown under red light, whereas in the dark plantlet length, seed vigor, seed germination, dry weight and pigment contents were decreased. Under blue and white lights, the main studied morphological parameters were decreased or close to red light. The cotyledons were undeveloped under dark conditions, likely due to the reduced content of photosynthetic pigments, which agrees with the low transcript levels of genes encoding protochlorophyllide oxidoreductase (*PORA*) and phytoene synthase (*PSY*). The transcript levels of a number of genes involved in phytohormone biosynthesis and signaling, such as *GA3ox*, *RRa*, *KAO* and *JazA*, were enhanced under red light, unlike under dark conditions. We suggest that the observed phenomena of red light are the most important for the germination of the plantlets and may be based on earlier and enhanced expression of auxin, cytokinin, gibberellin and jasmonate signaling genes activated by corresponding photoreceptors. The obtained results may help to improve reforestation technology; however, this problem needs further study.

## 1. Introduction

Seed germination occurs when provided with moisture and oxygen, suitable temperature and light conditions. The seeds of most plants germinate in the dark and do not need additional lighting [1]. Along with this, there are plants whose seeds need additional light during germination, as well as plants that are indifferent to the light regime. The size of the seed and, as a result, the size of the endosperm indicate that the seeds may be located deeper into the ground, which allows the plantlet to maintain a heterotrophic type of nutrition for a longer time [2]. Smaller seeds, especially weeds, on the contrary, perceive the ratio of red light (RL) to far red light (FRL) in light spectra, which allows them to germinate in the most favorable conditions in the absence of shading by neighboring plants [3]. Such plants tend to have a small endosperm and should begin photosynthesis as quickly as possible [4]. In seeds after imbibition, photoreceptors, such as phytochromes and cryptochromes, are activated, which affect the transition of the plantlet from heterotrophic to autotrophic nutrition [4]. Light signaling is mediated by RL photoreceptors—phytochromes and blue light (BL)—cryptochromes (CRY) and phototropins (PHOT) [5]. Phytochromes act as biological light switches by undergoing a reversible phototransformation from the inactive form of the Pr phytochrome to the active Pfr in response to RL and vice versa to the transition of the active form of Pfr to the inactive form in response to RL [6,7]. The inactive form of Pr localizes to the cytosol, while the RL-activated form of Pfr migrates to the nucleus [7]. In the nucleus, phytochromes activate various responses to light, including seed germination, by modulating the activity of various phytochrome-interacting proteins [8].

In *Arabidopsis thaliana* plants, five phytochromes (PHYA–PHYE) regulate different responses to light [9]. PHYA is the main phytochrome that promotes seed germination in response to very low irradiance of RL, while PHYB is the main phytochrome that promotes seed germination in response to low irradiance of RL [10]. Gymnosperms have a special system of phytochromes, since receptors of RL are represented by PHYN, PHYO and PHYP orthologues. In addition, they have the ability to synthesize chlorophyll in the dark due to the presence of three genes for light-independent protochlorophyllide oxidoreductases L, N and B (*chlL*, *chlN* and *chlB*), which are involved in the light-independent reduction of protochlorophyllide to chlorophyllide. On the one hand, exogenous hormones may induce plant growth by acting as mediators in light signal transduction processes [11]. Light can regulate the metabolism of various hormones and hormonal signaling genes. For example, it is well known that not only PHYs but also separately gibberellic acid (GA) and abscisic acid (ABA) modulate seed germination and that all the factors interact with each other [12,13]. Thus, RL via PHYA affects the metabolic pathways of gibberellins and indoleacetic acid [14], so the ratio of phytohormones plays a decisive role in these processes, mediating the transition to subsequent stages of ontogeny.

Phytochromes promote seed germination by decreasing the level of DELLA proteins through transcriptional repression of two DELLA genes and by increasing the level of bioactive GA, which activates the degradation of DELLA proteins [15]. Phytochromes also partially promote seed germination due to ABA antagonistic crosstalk. Light reduces ABA levels in seeds [7,16], primarily due to the repression of ABA biosynthesis genes (*ABA deficient* (*ABA1*) and *Nine-Cis-Epoxycarotenoid Dioxygenase 6* (*NCED6* and *NCED9*)) [17]. It has been shown using *phyb* mutants that light is able to repress ABA signaling not only through changes in its metabolism but also by repressing ABA-positive signaling genes [18]. Phytochromes regulate hormone metabolism and seed signaling via phytochrome-interacting factors (PIFs) [17], which inhibit seed germination [19]. Under dark conditions, PIF transcription factors (TFs) activate the expression of genes involved in ABA biosynthesis (*ABA1*, *NCED6* and *NCED9*) and inhibit the expression of genes involved in GA biosynthesis (*GA2ox2*). As a result, seeds have low levels of GA and high levels of ABA [17]. ABA inhibits seed germination, while GA promotes seed germination [20]. In addition to ABA and GA, other hormones (ethylene and brassinosteroids) stimulate or inhibit (auxins and cytokinins) seed germination [21]. Target genes for PIFs include genes encoding various hormone-related transcriptional regulators, such as Auxin response factor 18 (*ARF18*) and jasmonate ZIM-domain protein 1 (*JAZ1*). These results suggest that PIFs are involved in seed germination not only by regulating ABA and GA signaling but also by coordinating the signaling of other hormones.

In *A. thaliana*, light signaling involving phytochromes affects seed dormancy during both maturation and germination [22]. The dormant seeds of the *Brachypodium distachyon bd18-1* mutant show a photoreversible effect of RL/FRL on seed dormancy, suggesting a role for PHYB or possibly PHYC in seed dormancy disruption [23]. Our understanding of the role of PHY in dormancy/germination regulation in conifers is very limited due to the lack of mutants.

It is known that monocot seed germination can be inhibited by BL [24]. There is evidence linking BL CRYs perception to photoinhibition of germination in dormant monocot seeds [25,26]. In dormant barley seeds, green light partially abolishes the inhibitory effect of BL on germination, implying a role for CRY rather than PHOT in the light response [26,27]. Further studies showed that the overexpression of CRY in *A. thaliana* seeds resulted in increased sensitivity to ABA during germination compared to wild type. In *A. thaliana*, BL has been shown to partially inhibit germination, but this effect depends on PHYB rather than CRY1 [26].

Thus, plants have cross-signaling between different photoreceptors, which leads to the complex activation of biochemical processes during seed germination. Seeds are also a convenient system for estimating the amount of protein from their transcripts because in unimbibed seeds, transcripts of many genes are practically absent. The importance of RL and FRL wavelengths for the germination and growth of pine seeds has been shown in previous studies [28,29,30]; however, it is still not completely clear which molecular mechanisms of mutual gene expression regulation are involved in these processes. The role of blue light in the photoregulation of coniferous seed germination has also not been sufficiently studied.

Given the economic importance of Scots pine, the aim of this work was to study the effect of light on the germination of seeds, as well as on morphophysiological parameters, photosynthetic pigment contents and the transcript levels of photoreceptor genes, the main components of light and hormonal signaling. We wanted to establish which spectrum of light is able to maximize the activation of Scots Pine seed germination, as well as which photoreceptor genes, transcription factors, and hormonal signaling genes underlie this activation. We also tested the hypothesis that other hormones, besides ABA and GA, may be involved in the germination of Scots pine seeds.

## 2. Results

### 2.1. Seed Germination and Development of Plantlets

The radicle appearance at the early stage of seed germination was noted on the 3rd day after sowing under white fluorescent light (WFL) and white light (WL), on the 4th day under BL and RL, on the 5th day under FRL and only on the 6th day in the dark. As a consequence, the highest values of seed vigor were noted for WFL, WL and RL, while the minimum was observed in the dark. However, there were no differences in seed germination under different light qualities. At the same time, the minimum seed vigor and lowest seed germination were noted for seeds in the dark (Table 1).

As a consequence of rapid germination under WFL, plantlet weights were highest on the 6th day after seed germination, while comparable plantlet weights were achieved on the 7th day under BL and on the 9th day under RL and WL (Figure 1). The plantlet weight under FRL on the 9th day was 15.5% lower than that under WFL. Under WFL, the weight of plantlets increased by 3.1-fold during the experiment; in the dark, the increase in plantlet weight was only 41.9% (Figure 1).

In plantlets growing under WFL, release from the seed coat began on the 6th day of the experiment; while in plantlets growing under WL, BL and RL, it began on the 8th day; and under FRL, it began on the 9th day of the experiment. In the dark, plantlets maintained the seed coat until the end of the experiment (Figure 2).

RL and BL led to the formation of more developed plantlets. Therefore, on the 9th day of the experiment, the length and dry weight of plantlets grown under RL and BL were the highest compared to other variants (Table 2). At the same time, RL led to the development of a longer root and cotyledons, while BL and FRLs increased the hypocotyl length (Table 2). Notably, plantlets grown under FRL had the highest water content (Table 2).

### 2.2. Content of Photosynthetic Pigments

The maximum content of photosynthetic pigments in plantlets was observed on the 9th day of the experiment under RL. At the same time, the chlorophyll *a* (Chl *a*) and chlorophyll *b* (Chl *b*) contents in plantlets grown under RL exceeded their contents in the plantlets of all other variants. At the same time, under WFL, the pigment contents reached the maximum level on the 5th day of the experiment and did not change further. Under FRL, pigment accumulation was significantly reduced compared to that under the other light treatments. The minimum content of pigments was typical for plantlets developing in the dark (Figure 3). No trends in Chl a/b ratios were observed during the experiment; however, the ratio of carotenoids (Car) to the sum of chlorophylls *a* and *b* decreased as plantlets developed.

### 2.3. Transcript Levels of Photoreceptor Genes and TFs

On WFL and WL, the maximum level of the *PHYP* gene transcripts was noted at the beginning of seed germination, but after one day, they sharply decreased and were maintained at this level throughout the experiment. In contrast, an increase in the level of the transcripts of the *PHYP* gene was observed in RL during the experiment. For BL, the maximum level of the *PHYP* gene transcripts was noted on the 5th day of the experiment, while two peaks of expression were characteristic of FRL: on the 0th and 5th days. In the dark, the maximum level of the *PHYP* gene transcripts was observed on the 7th day of the experiment (Figure 4, Appendix A).

The maximum level of the *PHYO* gene transcripts was noted in plantlets under FRL. In addition, the greatest changes in the level of the *PHYO* gene transcripts were characteristic of plantlets grown in the dark (Figure 4).

In almost all variants, with the exception of darkness, the maximum level of the *PHYN* gene transcripts was observed at the beginning of seed germination and decreased during the experiment (Figure 4).

During the experiment on WFL, WL and BL, several maxima in the content of *CRY1* gene transcripts were noted, while minimal changes in the content of the transcripts were noted under FRL (Figure 4).

The maximum levels of the *CRY2* gene transcripts were noted in RL and BL; however, RL was characterized by a more pronounced and prolonged increase in the level of the *CRY2* gene transcripts (Figure 4).

The most pronounced increase in the levels of *HY5* gene transcripts was noted in the dark and on WFL. At the same time, the level of the transcripts of this gene on RL did not change during the entire experiment (Figure 4).

In the dark, the level of the *PIF3* gene transcripts strongly increased during the experiment, reaching a maximum on the 7th day. Similar but less pronounced trends in the increase in the content of the transcripts of this gene were also noted in the variants FRL, RL and BL. In contrast, WFL was characterized by a decrease in the level of the *PIF3* gene transcripts during the experiment, and no changes in the content of the transcripts were noted in WL (Figure 4).

**Figure 3 ijms-24-02063-f003:**
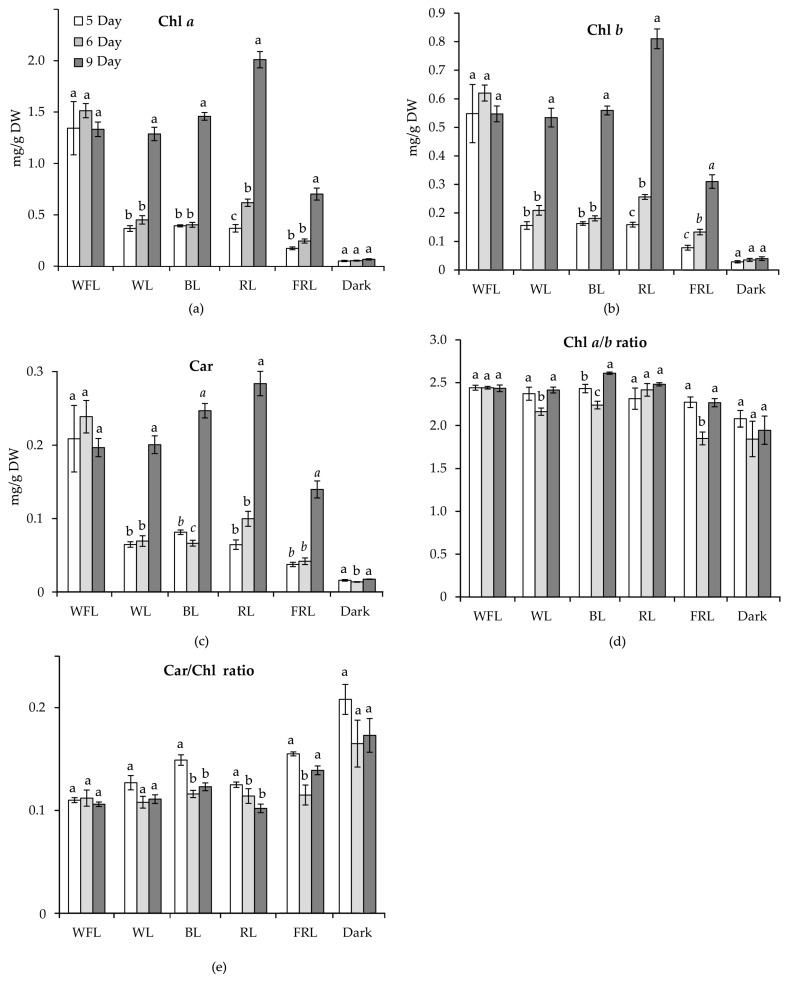
Content of photosynthetic pigments, as well as pigment ratio of Scots pine plantlets, on the 3rd, 5th and 9th day of the experiment under different light qualities. The contents of chlorophyll *a*, *b* and carotenoids are expressed in mg/g dry weight: (**a**) Chl *a*; (**b**) Chl *b*; (**c**) Car; (**d**) Chl *a*/*b* ratio; (**e**) Car/Chl. Different letters denote statistically significant differences in the means at *p* < 0.05 (ANOVA followed by Duncan’s method) between the experimental treatments. Different italic letters denote statistically significant differences in the means at *p* < 0.05 (Kruskal–Wallis ANOVA of the ranks followed by the Student–Newman–Keuls post hoc test) between the experimental treatments.

**Figure 4 ijms-24-02063-f004:**
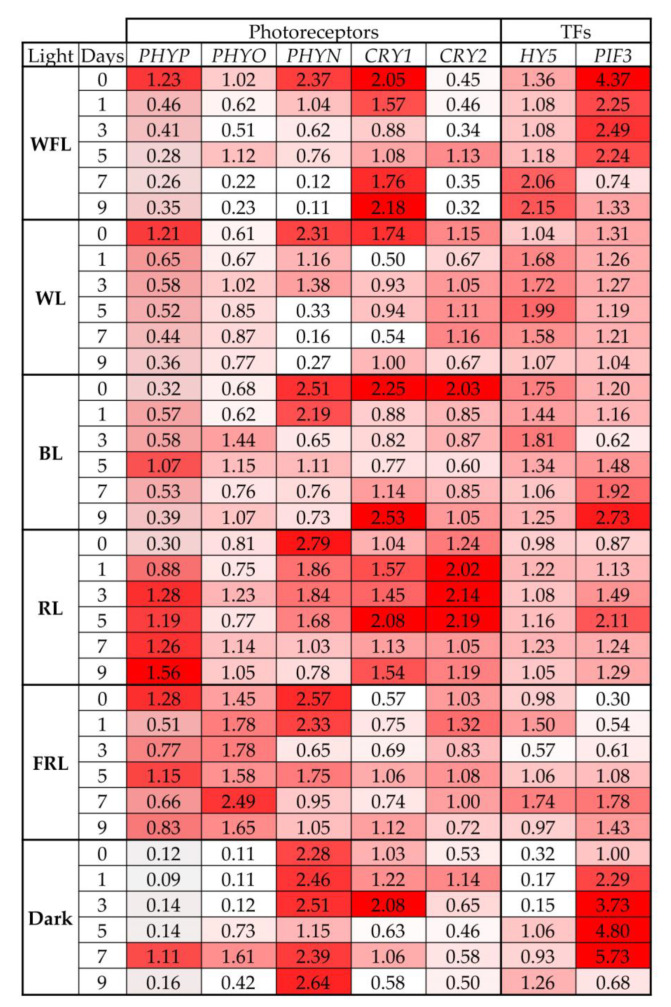
Transcript levels of main photoreceptors (*PHYP, PHYO, PHYN, CRY1, CRY2*) and transcription factors involved in light signaling (*HY5, PIF3*) in Scots pine plantlets under different light quality and dark conditions on days 0, 1, 3, 5, 7 and 9 of the experiment. The color scale rules: increase in expression by at least two times (dark red light) relative to the average value of the expression of the specified gene in column.

### 2.4. Transcript Levels of Genes for Photosystems and Pigment Biosynthesis

The transcript levels of *LHCa puc* and *LHCb2* genes increased during the experiment in all experimental variants in a coordinated manner, reaching a maximum on the 9th day (Figure 5, Appendix A).

The most pronounced changes in the levels of the *rbcL* gene transcripts were characteristic of plantlets on FRL and RL. Under WFL and WL, the maximum levels of the *rbcL* gene transcripts were noted on the 9th day, while in the dark, they were noted on the 1st day of the experiment (Figure 5).

The maximum levels of the *PORA* gene transcripts were noted under RL and WL. However, unlike other variants, the level of the *PORA* gene transcripts in the dark gradually increased during the experiment, and on FRL, the expression intensity of this gene did not change at all (Figure 5).

PSY gene transcripts in all variants, except for darkness, increased during the experiment and reached their maximum values at WL, BL and RL. At the same time, for seedlings on RL, the maximum content of the transcripts of this gene was observed much earlier than in other variants on the 5th day of the experiment, which could ensure the accumulation of an increased content of carotenoids on the 9th day of the experiment (Figure 5, Appendix A).

### 2.5. Expression of Phytohormone Biosynthesis and Signaling Genes

Changes in the levels of the *GA3ox1* gene transcripts were typical only for WFL, WL, BL and RL at the end of the experiment. At the same time, the maximum increase in the *GA3ox1* gene transcripts was noted in WFL (Figure 6, Appendix A).

The maximum levels of the *GA3ox2* gene transcripts were observed in plantlets grown under WL and RL. At the same time, under WFL and BL, there were two peaks in the content of the transcripts of this gene. In the dark and FRL, the content of *GA3ox2* gene transcripts did not change (Figure 6).

The maximum increase in the content of the *KAO1* gene transcripts was observed at the beginning of the experiment in plantlets grown in the dark and under FRL. In other variants, the content of the *KAO1* gene transcripts increased towards the end of the experiment. The levels of the transcripts of the *KAO2* gene increased during the experiment in all variants, except for darkness and FRL, where the content of the transcripts of this gene did not change (Figure 6).

In the dark and under FRL and RL, the transcript level of the *NCED* gene did not change during the experiment. At the same time, a significant increase in the content of gene transcripts was observed on WFL and BL: from day 1 on WFL and from day 5 on BL. Under WL, an increase in the content of the transcripts of the *NCED* gene was only observed at the end of the experiment (Figure 6, Appendix A).

A sharp decrease in the level of the *PYR* gene transcripts from the 1st day of the experiment was noted under BL and RL. A similar but less pronounced decrease in the level of the *PYR* gene transcripts was noted under WFL. At the same time, for WL, FRL and dark, several peaks were noted in the content of the *PYR* gene transcripts (Figure 6).

In all variants of the experiment, an increase in the transcript levels of the *CCH1* gene was noted. The maximum level of the transcripts of this gene was typical for plants grown in the dark. At the same time, the strongest increase in the content of the *CCH1* gene transcripts during the experiment was observed in WL and FRL (Figure 6).

The high levels of the *AUX/IAA* gene transcripts were characteristic of seedlings developing in the dark, FRL and RL (Figure 6). The most pronounced increase in transcript levels was observed on the 5th day of the experiment under WFL and on the 1st and 3rd days of the experiment under RL. In addition, in the dark, the transcript level of this gene remained high throughout the experiment (Figure 6, Appendix A).

The highest content of the *RRa* gene transcripts was characteristic of BL throughout the experiment. However, the most pronounced increase in the level of the transcripts was observed under WFL and WL. The FRL was characterized by a maximum content of the *RRa* gene transcripts on the 1st day, after which the level of the transcripts of this gene decreased (Figure 6, Appendix A).

The most significant increase in the content of the *PR1* gene transcripts was noted under RL on the 1st day of the experiment, while in most other variants, it was noted by the end of the experiment.

The maximum level of the *JazA* gene transcripts was observed in plantlets grown under RL. However, the greatest increase in transcripts was observed in plantlets in the dark at the beginning of the experiment and under BL on the 9th day of the experiment.

## 3. Discussion

The germination of pine seeds depends on illumination and temperature and is species specific. The pyrogenic species *P. halepensis* and *P. brutia* are characterized by rather low germination in the dark at 20 °C; in addition, their germination is photosensitive and activated by WL but inhibited by FRL. At the same time, the germination of *P. pinea* and *P. nigra* seeds is independent of light conditions, while seeds of *P. heldreichii* and *P. sylvestris* require light and/or stratification for germination [31]. Pine seeds have a relatively large endosperm, which allows them to germinate in the soil, where sunlight does not penetrate. At the same time, mature seeds are dispersed by wind and germinate on the surface of the soil after imbibition, which allows sunlight to penetrate into the seed. Under the forest canopy where the seeds germinate, there is light with a reduced RL/FRL ratio, which, according to previous studies, may negatively affect seed germination [31]. Gymnosperms respond differently to light intensity and the RL/FRL ratio, although some aspects of the response may be conservative. The conditions of light and temperature in which the mother plant is vegetated can also affect the properties of the seeds because light and temperature signals can accumulate as chemical signaling molecules in seeds [32].

In well-lighted areas at noon, the RL/FRL ratio is reported to be 1.2 [33], whereas in plant shade, RL is largely absorbed by chlorophyll and other leaf pigments, while FRL is transmitted or reflected, resulting in a reduction in the RL/FRL ratio to 0.2–0.8 [33]. Similarly, the decrease in the RL/FRL ratio occurs during the twilight hours mainly due to the longer path of light through the ozone layer, which absorbs in the red-green region of the spectrum, resulting in a relative increase in blue and FR light. Fernbach and Mohr [34] showed the absence of a hypocotyl growth response in *P. sylvestris* seedlings to high illumination (HIR) (i.e., no inhibition of hypocotyl growth at FRL), while the presence of HIR at FRL is a common feature of angiosperms. Thus, gymnosperms react differently to light quality compared to angiosperms. In gymnosperms, as in angiosperms, the degree of vegetative response to shading differs between species depending on their level of shading tolerance [35]. Shade tolerance is the ability of a particular species to efficiently change its morphology to adapt and respond to low light intensity and a low RL/FRL ratio, in addition to changing carbon distribution from photosynthetic activity to elongation [36]. In *A. thaliana*, shadow avoidance syndrome is well characterized, and this response primarily involves PHYA and PHYB, as well as genes required for the integration of light and hormonal signaling pathways [37]. FRL reduces phyB activity, which increases the mRNA levels of the major helix–loop–helix (bHLH) transcription factors, such as phytochrome interaction factors (PIF) [38]. PIFs regulate many gene-encoding metabolic enzymes and phytohormone signaling pathways [39]. In our experiments, we observed an increase in the level of *PHYP* gene transcripts under RL throughout the experiment, while the other two phytochromes, *PHYO* and *PHYN* genes, were activated under FRL and darkness conditions, respectively (Figure 4). This indicates the importance of PHYP in the germination of Scots pine seeds. The level of *PIF3* transcripts remained at the maximum level in the dark, while under WFL, *PIF3* expression was also high but significantly decreased to the end of the experiment (Figure 4). PIF TFs inhibit seed germination and, together with phytochromes, are involved in the coordination of hormonal signals that affect seed germination [7,40]. RL phytochrome activation promotes PIF degradation, leading to seed germination [7]. In addition to light reactions, PIFs are involved in the regulation of the transcription of gibberellic acid signaling genes, as well as other components of hormonal signaling systems. This is additionally confirmed by the results of our experiments, according to which the highest activation of the transcription of the GA biosynthesis and signaling genes (*GA3ox2* and *KAO1*) was observed under RL (Figure 6). In addition, PIFs are involved in the regulation of the transcription of ABA signaling genes [33]. In our experiment, ABA signaling genes were activated in the dark and remained at high levels until the end of the experiment (Figure 6).

Hormones, such as auxins, cytokinins, brassinosteroids, ethylene and jasmonic acid, regulate seed germination. It can be assumed that hormone-related transcriptional regulators influence hormonal signaling during seed germination. We assume that RL has a particular effect on the germination of Scots pine seeds. In addition to the maximum values of the root length and the earlier onset of photosynthetic pigment accumulation, we observed the activation of a complex of hormonal signaling genes, such as *AUX/IAA* (auxins), *HPT1* and *RRA* (cytokinins), and *JazA* (jasmonates) (Figure 6). Since these processes must occur in all cells [20], seeds can use various hormonal signals to coordinate necessary physiological and developmental processes during seed germination. The germination process includes the mobilization of stored nutrients, the elongation of root initial cells and the rupture of both the endosperm and seed coat. In addition, seeds must trigger biochemical and cellular processes to activate cell proliferation. The resulting effect of light-hormonal regulation is the transition from heterotrophic scotomorphogenesis to autotrophic photomorphogenesis and the development of various organs. In our studies, FRL and BL had the greatest effect on hypocotyl length, while the dry weight of plantlets was maximal under BL and RL (Figure 1, Table 2). At the same time, the maximum percentage of water content was observed under FRL (Table 2). This indicates that, under FRL, Scots pine plantlets develop according to the type of shade avoidance syndrome, i.e., did not grow by cell division but rather by cell expansion. This is a typical reaction that is characteristic of dicotyledonous herbaceous plants [41]. At the same time, the length of cotyledons and roots was maximal under RL. The effect of RL on the root system of 6-week-old Scots pine seedlings was shown in our previous study [42].

Simultaneously, with the inactivation of negative factors of photomorphogenesis, we observed the activation of the level of the transcripts of positive photomorphogenesis regulators, such as HY5. An increase in the level of the transcripts of the TF *HY5 gene* under WFL on the 7th and 9th days of the experiment, as well as under WL, probably indicates that RL together with BL can affect the transcription of this gene (Figure 4). In turn, HY5, which promotes photomorphogenesis in the light, interacts with photoreceptors and directly affects the transcription of light-dependent genes induced by RL. In addition, HY5 is also involved in BL signaling, suggesting that it is involved in cryptochrome (CRY1 and CRY2) signaling [43]. In our experiment, the highest activation of blue wavelength photoreceptors was observed under RL, while the highest expression of *HY5* was observed under BL and WL (Figure 4). At the same time, HY5 regulates the ABA and auxin signaling pathways and is involved in the suppression of hypocotyl cell elongation during photomorphogenesis [43]. On the other hand, hormonal cross-signaling was recently shown in *A. thaliana* in which both auxins and jasmonates are involved [44]. We assume that a similar mechanism is possible for pine trees under RL conditions.

Ultimately, the key stage in seed germination is the transition to an autotrophic type of nutrition, in which pigment biosynthesis plays a decisive role, as well as the formation of chlorophyll-binding proteins in chloroplast membranes [40]. In our experiments, the greatest activation of the level of the transcription of the chlorophyll-binding protein genes (*LHCa1*, *LHCb2*) was observed on WFL and WL (Figure 5). At the same time, the expression of the *PORA* chlorophyll biosynthesis gene under RL was observed on the 3rd day of the experiment—earlier than in other variants (Figure 5). In addition, the expression of *PSY*, the main gene of carotenoid biosynthesis, was also increased under RL before than in other studied variants. These results are additionally confirmed by the accumulation of photosynthetic pigments by the 9th day of the experiment under RL, which was preceded by the accumulation of the transcripts of the corresponding gene.

## 4. Materials and Methods

### 4.1. Experimental Design

Seeds of Scots pine (*Pinus sylvestris* L.) were provided by the Training and Experimental Forestry Enterprise of the Bryansk State Technological University of Engineering (Bryansk, Russia) and were collected in the Bryansk region from high-productive pine stands in complex forest types. The seeds were stored in a refrigerator at +4 C in a tightly closed glass bottle. The seeds were sown into plastic boxes (15 × 15 cm) filled with 1% agar in a layer 10 cm thick. Immediately after sowing, boxes with seeds were placed in the climatic chamber under red (maxima of 660 nm) (RL), blue (maxima of 450 nm) (BL), white light (maxima of 450 nm and 575 nm) (WL) LEDs, fluorescent lamps (58 W/33–640, white fluorescent lamps (Philips, Pila, Poland) (WFL), 130 ± 10 µmol (photons) m^−2^ s^−1^ [42]) and under far red (maxima of 730 nm) (FR) LED, as well as under dark conditions (Figure 7).

Boxes with seeds were abundantly sprayed with distilled water directly in the climatic chambers, after which the seeds were allowed to germinate at 22 ± 2 °C for a 16 h photoperiod. Thus, seed imbibition and germination occurred directly under the specified light spectra. To maintain the necessary moisture in the boxes, the seeds were sprayed daily with distilled water. Seed germination was assessed several times over a period of 15 days, while seed vigor was assessed over a period of 7 days after seed sowing. The excess of the radicle length over the length of the seed was the criterion of the germinated seed [45,46].

The plantlets with seed coats were collected at the beginning of germination (0), and on the 1st, 2nd, 3rd, 4th, 5th, 6th, 7th, 8th and 9th days after massive seed germination in all variants. Of the plantlets collected at each time point, 5–10 plantlets were grouped into composite samples. Each composite sample was treated as a biological replicate.

### 4.2. Fresh Weight and Water Content

The fresh weights of the composite samples were determined daily with an accuracy of 1 mg using an analytical balance (Scout Pro SPU123, Ohaus Corporation, Parsippany, NJ, USA). The fresh weight of individual plantlets was calculated by dividing the weight of the composite sample by the number of plantlets in the composite sample. The dry weights of the samples on the 5th, 6th and 9th days of the experiment were determined using an analytical balance (AB54-S, Mettler Toledo, Greifensee, Switzerland) with an accuracy of 0.1 mg after drying the samples to a constant weight at 70 °C. The water content in plantlets was expressed as a percentage of their fresh weight [47]. On the 5th, 6th and 9th days of the experiment, the water content was calculated for plantlets with seed coats, and on the 9th day of the experiment, the water content was calculated for plantlets without seed coats separately.

### 4.3. Morphometric Parameters

To determine the linear dimensions of 9-day-old plantlets, they were laid out individually on the glass of an Epson Perfection V500 Photo flatbed scanner (Epson, Suwa, Nagano, Japan) and scanned at a resolution of 800 dpi. The public domain software ImageJ v.1.49 (NIH; http://rsb.info.nih.gov/ij, accessed on 30 January 2021) was used to measure the lengths of the plantlets (root, hypocotyl, cotyledons, whole plantlet) to an accuracy of 0.1 mm.

### 4.4. Determination of Photosynthetic Pigments

The chlorophyll *a* (Chl *a*), *b* (Chl *b*) and carotenoid (Car) contents were determined using the Lichtenthaler method [48] in plantlets on the 5th, 6th and 9th days of the experiment. The samples were triturated with 80% acetone in the dark. The absorbance of the centrifuged samples was measured with a Genesys 10 UV–Vis spectrophotometer (Thermo Fisher Scientific, USA) at wavelengths of 470, 646 and 663 nm. The content of the photosynthetic pigments was determined using the Lichtenthaler formulas:Chl *a* = 12.25 × A_663_ − 2.79 × A_646_,(1)
Chl *b* = 21.50 × A_646_ − 5.10 × A_663_,(2)
Car = (1000 × A_470_ − 1.82 × Chl *a* − 85.02 × Chl *b*)/198,(3)

### 4.5. RNA Extraction and Quantitative RT–PCR

RNA isolation was performed according to a method based on aurintricarboxylic acid [49,50]. The cDNA synthesis and qRT–PCR analysis of gene expression patterns were performed according to Pashkovskiy et al. 2021 [42]. The list of gene-specific primers is given in Appendix A. The transcript levels were normalized to the expression of the *Actin1* gene.

### 4.6. Statistical Analysis

The number of biological replicates in the determination of fresh weight of the plantlets ranged from 3 to 6. The number of biological replicates assessed in the determination of pigment content and water content was 4. The number of biological replicates assessed in the determination of the morphometric parameters of 9-day-old plantlets was 20. Three biological replicates were performed to determine the transcript levels of genes.

Statistical analyses of data were performed using SigmaPlot 12.3 (Systat Software, San Jose, CA, USA) with one-way analysis of variance (ANOVA) followed by Duncan’s method for normally distributed data (significant differences are denoted by different normal letters) and Kruskal–Wallis one-way ANOVA on ranks followed by the Student–Newman–Keuls post hoc test for non-normally distributed data and data with unequal variance (significant differences are denoted by different italic letters). The values presented in the tables and figures are the arithmetic means ± standard errors.

## 5. Conclusions

Thus, the germination of Scots pine seeds depends little on the spectral composition of light, but in the dark, germination noticeably deteriorates.

Unlike germination, the development of plantlets is related to the quality of light. The highest rates of dry weight accumulation, root length and photosynthetic pigment content were observed in plantlets grown under RL. Apparently, the accumulation of pigments under RL is due to earlier and more intense expression of the *PORA* and *PSY* genes of enzymes responsible for the biosynthesis of these pigments.

We suggest that the activation of plant development under RL and other regions of the spectrum is achieved through coordination between the expression of the *PHYP, PHYN, CRY2, PIF3, HY5* genes and hormone signaling genes, in particular, *RRa, GA3ox1* and *GA3ox2, KAO1* and *KAO2*.

Thus, our study examines the genetic basis for the activating action of RL on Scots pine seed germination. Additionally, based on the data obtained, it was concluded that it is important to grow Scots pine seedlings under artificial lighting containing a sufficient amount of RL.

## Figures and Tables

**Figure 1 ijms-24-02063-f001:**
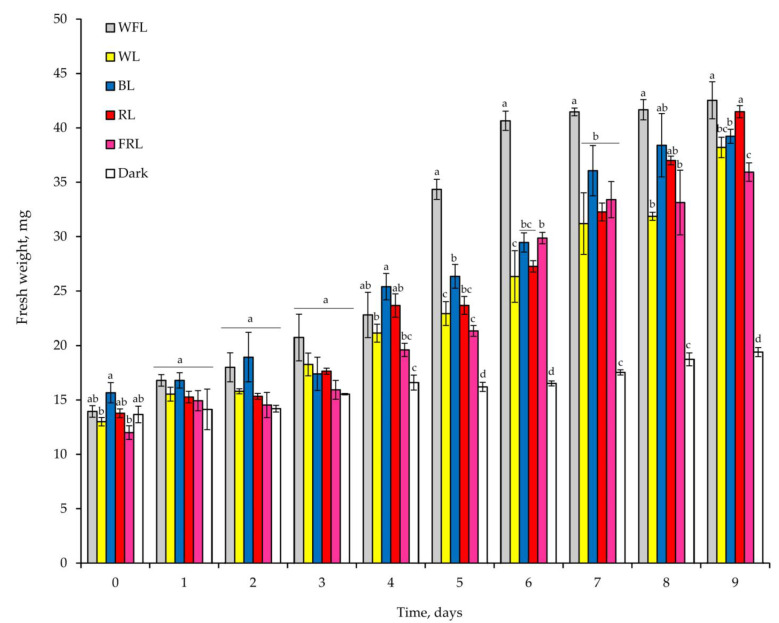
Effect of light quality on the fresh weight of Scots pine plantlets with seed coats during the experiment. Different letters denote statistically significant differences in the means at *p* < 0.05 (one-way ANOVA followed by Duncan’s post hoc test) between the experimental treatments within each day of observation, *n* = 3–6.

**Figure 2 ijms-24-02063-f002:**
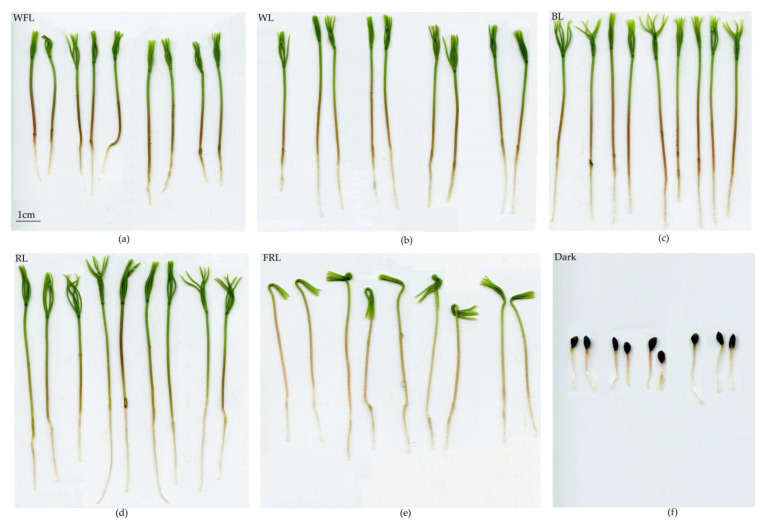
Nine-day-old Scots pine plantlets grown under different light regimes: (**a**) WFL; (**b**) WL; (**c**) BL; (**d**) RL; (**e**) FRL; (**f**) Dark.

**Figure 5 ijms-24-02063-f005:**
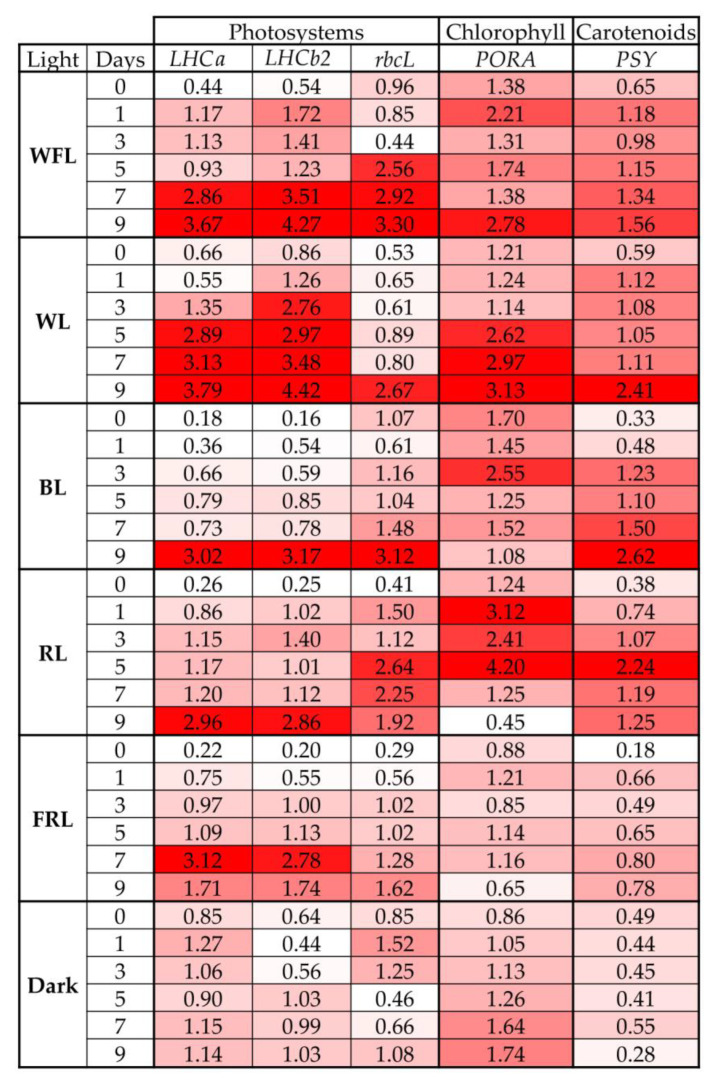
Transcript levels of main genes of light harvesting complex of Chl-*a*/*b*-binding proteins (*LHCa*, *LHCb2*), and large subunit of RubisCo (*rbcL*), chlorophyll biosynthesis gene protochlorophyllide oxidoreductase (*PORA*), main gene of carotenoids biosynthesis phytoene synthase (*PSY*) in Scots pine plantlets under different light quality and dark on the 0, 1, 3, 5, 7, 9 days of experiment. The color scale rules: increase in expression by at least two times (dark red light) relative to the average value of the expression of the specified gene in column.

**Figure 6 ijms-24-02063-f006:**
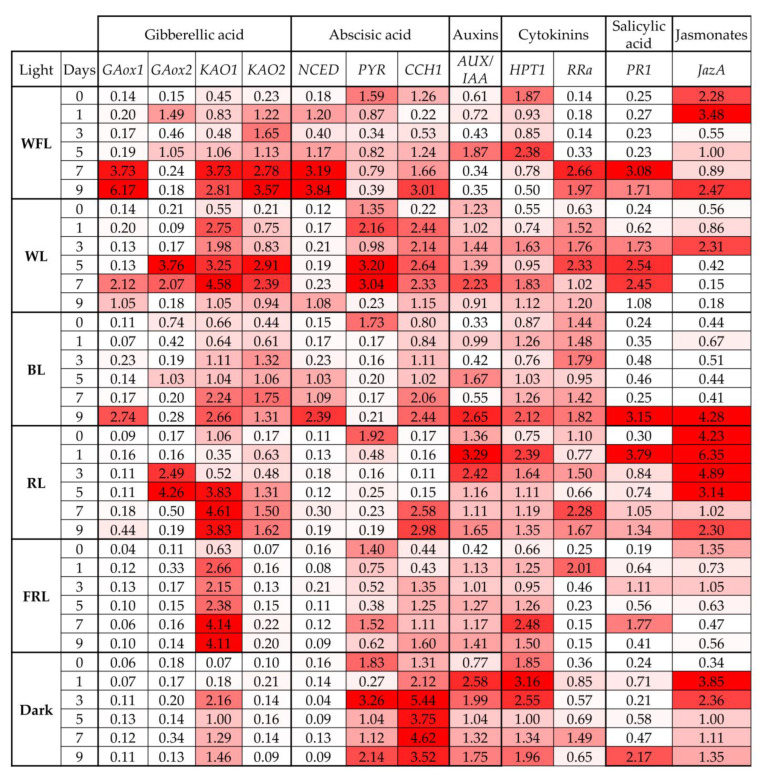
Transcript levels of genes of gibberellic acid biosynthesis and signaling (*GA3ox1, GA3ox2, KAO1, KAO2*), as well as ABA biosynthesis and signaling (*NCED, PYR, CCH1*), main genes of auxin signaling (*AUX/IAA*), cytokinin signaling (*HPT1, RRa*), salicylic acid signaling (*PR1*) jasmonic acid signaling (*JAZa*), in Scots pine plantlets under different light quality and dark conditions on days 0, 1, 3, 5, 7 and 9 of the experiment. The color scale rules: increase in expression by at least two times (dark red light) relative to the average value of the expression of the specified gene in column.

**Figure 7 ijms-24-02063-f007:**
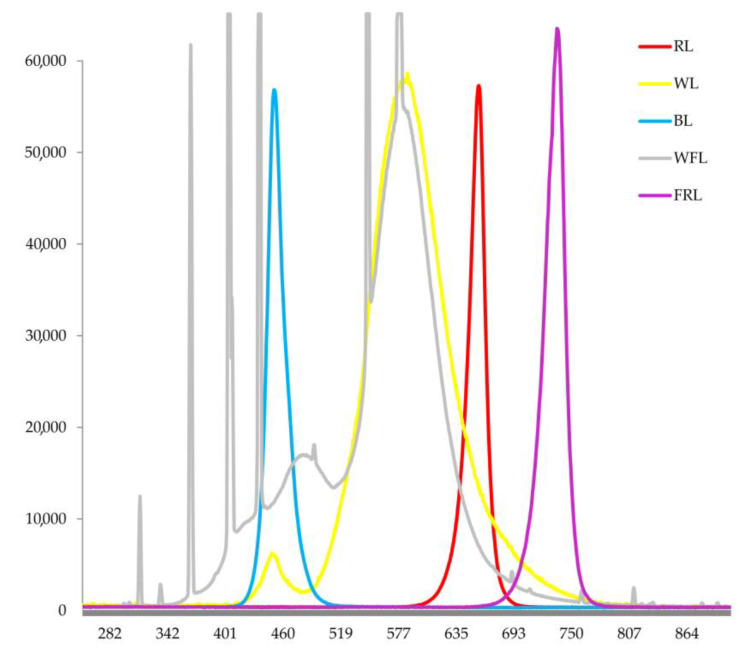
Emission spectra of various light sources used in the experiment.

**Table 1 ijms-24-02063-t001:** Effect of light quality on the vigor and germination of Scots pine seeds.

Light	Seed Vigor, %	Seed Germination, %
WFL	73.2 ± 3.3 a	80.6 ± 4.9 a
WL	65.6 ± 4.7 ab	78.4 ± 5.8 a
BL	55.7 ± 2.8 b	80.2 ± 4.7 a
RL	58.9 ± 5.0 ab	80.8 ± 3.3 a
FRL	48.1 ± 7.4 b	81.6 ± 6.6 a
Dark	11.4 ± 4.3 c	56.6 ± 3.9 b

Different letters denote statistically significant differences in the means at *p* < 0.05 (ANOVA followed by Duncan’s method).

**Table 2 ijms-24-02063-t002:** Growth parameters of Scots pine plantlets on the 9th day of the experiment under different light and dark conditions.

Light	Plantlet Length cm	Root Length cm	Hypocotyl Length cm	Cotyledons Length, cm	Plantlet Dry Weight (without Seed Coat), mg	Water Content in Plantlet, %
WFL	5.62 ± 0.11 *c*	1.37 ± 0.06 c	2.98 ± 0.09 c	1.24 ± 0.04 c	3.78 ± 0.08 *ab*	86.40 ± 0.21 *b*
WL	6.58 ± 0.10 *b*	1.93 ± 0.08 b	3.47 ± 0.09 b	1.14 ± 0.03 c	3.86 ± 0.10 *ab*	87.10 ± 0.24 *b*
BL	7.27 ± 0.12 *a*	1.84 ± 0.08 b	4.03 ± 0.07 a	1.37 ± 0.04 b	4.43 ± 0.19 *a*	86.33 ± 0.10 *b*
RL	7.42 ± 0.19 *a*	2.29 ± 0.14 a	3.49 ± 0.11 b	1.59± 0.03 a	4.64 ± 0.16 *a*	86.60 ± 0.23 *b*
FRL	6.60 ± 0.19 *b*	1.75 ± 0.08 b	3.79± 0.13 a	1.02± 0.04 d	3.19 ± 0.09 *b*	88.68 ± 0.13 *a*
Dark	1.89 ± 0.09 *d*	1.42 ± 0.09 c	ND	ND	3.57 ± 0.37 *ab*	65.73 ± 2.08 *c*

Different letters denote statistically significant differences in the means at *p* < 0.05 (ANOVA followed by Duncan’s method). Different italic letters denote statistically significant differences in the means at *p* < 0.05 (Kruskal–Wallis ANOVA of the ranks followed by the Student–Newman—Keuls post hoc test).

## Data Availability

The datasets generated and/or analyzed during the current study are available from the corresponding author on reasonable request.

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
