# Peer review of "Influence of Light of Different Spectral Compositions on Growth Parameters, Photosynthetic Pigment Contents and Gene Expression in Scots Pine Plantlets"

_ijms, 2023, doi:10.3390/ijms24032063_

Round 1
Reviewer 1 Report
In the manuscript “Influence of Light Quality on the Growth Parameters, Photosynthetic Pigment Contents and Gene Expression in Scots Pine Plantlets”, the authors investigated morphological traits under different light quality, as well as the transcript levels of genes involved in photosynthesis and hormonal signaling of Scots pine plantlets. This manuscript will be of interested by photo biology researchers, when some issues should be clarified.
Comments:
1. Line 56-57, “PHYA is the main phytochrome that promotes seed germination in response to low irradiance of RL, while PHYB is the main phytochrome that promotes seed germination in response to low irradiance of RL”. Totally duplicated description for PHYA and PHYB.
2. Line 121-123, seed vigor is a complex trait and mainly affected by seed harvesting, processing, and storage conditions, how did seed vigor be measured in this manuscript?
3. Line 135, how many replicates and samples were used for measuring the fresh weight of Scots pine plantlets in Figure 1? Why the fresh weight at the beginning of germination (0day) varies a lot under different condition? A line chart should be better when comparing the influence between different light quality during the experiment.
4. Figure 2, the plantlet length varies a lot, especially in panel (e).
5. The color scale rules in Figure 5, 6 and 7 look different. It would be helpful to be much more explicit to describe them in the legend (for example, by row, by column, or by light conditions...), so that the gene expression levels under different conditions or time points can be clearly compared according to the cell color. In these experiments, it is more appropriate to use a two-color scale (for example, white-red) for sequential transcript values, because it is meaningless to use white to represent values in the middle.
6. Many genes are involved in phytohormone biosynthesis and signaling pathway. What are the criteria for selecting genes for qRT-PCR, are these listed genes representative? Why didn't authors check the expression of the genes mentioned in introduction section, such as ABA1 or ARF18?
Author Response
Reviewer 1
1. Line 56-57, “PHYA is the main phytochrome that promotes seed germination in response to very low irradiance of RL, while PHYB is the main phytochrome that promotes seed germination in response to low irradiance of RL”. Totally duplicated description for PHYA and PHYB.
Answer: Done
2. Line 121-123, seed vigor is a complex trait and mainly affected by seed harvesting, processing, and storage conditions, how did seed vigor be measured in this manuscript?
Answer:
Scots pine cones were collected in late autumn 2016 - winter 2017. According to the industry standard (Russian: ОСТ “otraslevoj standart” (OST) 56-23-75), seeds were removed from the cones in a cone dryer and stored in hermetically sealed glass bottles at +4 °C in the dark. Pine seeds from this seed lot were used until 2021 (see Zlobin et al., Photosynthesis Research, 2019; Pashkovskiy et al., Plant Physiology and Biochemistry, 2019; Ivanov et al., Environmental and Experimental Botany, 2019; Ivanov et al., Environmental Science and Pollution Research, 2021; Ivanov et al., Cells, 2022).
According to governmental standards (Russian: All-Union state standard (GovStandart), 13056.6-97. Seeds of trees and shrubs. Seed germination is the ability of seeds to form normally developed plantlets at a certain time; seed vigor (or germination energy) is the ability of seeds to germinate quickly in a certain period of time. For Scots pine, seed vigor was estimated after 7 days, and seed germination was estimated 15 days after seed soaking. Since this document exists only in Russian and is not available to most readers, the references to our previously published articles are done, in which time for seed vigor and seed germination indicated. In addition, this approach was also used in our previous works (with different seed lots of Scots pine seeds) not mentioned in this manuscript (see Ivanov et al., Plant Physiology and Biochemistry, 2016; Ivanov et al., Environmental Science and Pollution Research, 2016).
In section 4.1, necessary explanations were added.
3. Line 135, how many replicates and samples were used for measuring the fresh weight of Scots pine plantlets in Figure 1? Why the fresh weight at the beginning of germination (0day) varies a lot under different condition? A line chart should be better when comparing the influence between different light quality during the experiment.
Answer: We are grateful to the reviewer for the note. The number of biological replicates in the determination of fresh weight of the plantlets ranged from 3 to 6. At the beginning of germination (0 day), the number of replicates was 3. At this time point, statistically significant differences were only between BL and FRL variants (30.6%) and between BL and Dark variants (14.6%). Different letters in Figure 1 denote statistically significant differences in the means at p < 0.05 between the experimental treatments within each day of observation. These letters are not comparable between different light regimes. The necessary explanation was added to the figure caption. We changed Figure 1 and placed time on the x-axis to make it clearer. Unfortunately, a line chart is difficult to perceive, because there is no places for letters indicating significant difference.
4. Figure 2, the plantlet length varies a lot, especially in panel (e).
Answer: We are grateful to the reviewer for the remark. Figure 2 was given for illustrative purposes, and due to the lack of physiological meaning and clarity, we decided to exclude it from the revised version of the manuscript.
5. The color scale rules in Figure 5, 6 and 7 look different. It would be helpful to be much more explicit to describe them in the legend (for example, by row, by column, or by light conditions...), so that the gene expression levels under different conditions or time points can be clearly compared according to the cell color. In these experiments, it is more appropriate to use a two-color scale (for example, white-red) for sequential transcript values, because it is meaningless to use white to represent values in the middle.
Answer: We are grateful to the reviewer for a useful recommendation. We have corrected the figures with the two-color bar, and they are truly more understandable. We also improved the caption to the figures.
6. Many genes are involved in phytohormone biosynthesis and signaling pathway. What are the criteria for selecting genes for qRT-PCR, are these listed genes representative? Why didn't authors check the expression of the genes mentioned in introduction section, such as ABA1 or ARF18?
Answer: We tried to choose the key genes involved in hormonal signaling. Because we did not determine the content of phytohormones and in cases where transcription of hormone signaling genes is not activated, the content of specific hormones is less important. In addition, despite the presence of a constantly supplementing pine transcriptome, many genes have only predicted properties. In this regard, we were limited in the choice of genes.
Zeaxanthin epoxidase (ABA1) is not annotated in pine; instead, we studied the annotated: 9-cis-epoxycarotenoid dioxygenase genes, abscisic acid receptor PYR and ABA-binding protein Chloroplast Mg2+-chelatase/ABA Receptor.
Unfortunately pine does not have ARF18, it was necessary to study it by enumeration, instead we used the key auxin signaling gene AUX/IAA, which cannot be not involved in hormonal signaling during seed germination
Reviewer 2 Report
The problem of light-germinating and dark-germinating seeds can undoubtedly be important in the restoration of forest landscapes. An additional study of Phytochrome Interacting Factors (PIFs) on the example of Scots pine seeds will allow a deeper understanding of the mechanism of germination, its adaptation. The paper's subject matter corresponds to the journal's focus. The level of presentation of the manuscript provides sufficient understanding of the research process.
However, I have to give substantial recommendations:
1. Abstract
1.1. L16-17 – the wavelength value in nanometers is required for each region used in study.
1.2. It is necessary to add two sentences at the beginning of the abstract that answer the question: Why did you do it? (2 sentences, max 50 words). Briefly outline the problem you are tackling and why it is important.
1.3. It is necessary to add two sentences at the end of the abstract that answer the questions: "What do the results mean in practice? and What remains unresolved?"
2. The keywords should include the type of plant (Latin name) whose seeds were used for the study.
3. DOI must be specified in the references.
4. Introduction
4.1. It is necessary to focus more on the processes occurring under the influence of light in the seeds of gymnosperms, or to indicate more clearly the transition from the model plant – Arabidopsis – to Pinus sylvestris L.
4.2. It should be more clearly indicated why the species Pinus sylvestris L. was chosen, since the phrase "seeds are also a convenient system for estimating the amount of protein from their transcripts" can refer to any tree species.
4.3. The phrase "We assumed that the presence of RL in the light spectrum has the greatest stimulating effect on the germination and grow parameters of Scots pine seeds" requires correction, since there are diametrical studies of the effect of RL on the germination process of Scots pine seeds, and this hypothesis is not new. It should be clearer to highlight the originality of the research goal in comparison with existing studies that RL stimulates
(Qamaruddin and Tillberg, 1989. Rapid effects of red light on the isopentenyladenosine content in Scots pine seeds. Plant Physiol. 91:5-8;
Tillberg, 1992. Effect of light on abscisic acid content in photosensitive Scots pine (Pinus sylvestris L.) seed. Plant Growth Reg. 11:147-152).
or inhibits the germination process of Scots pine seeds
(Novikov, A.; Bartenev, I.; Podvigina, O.; Nechaeva, O.; Gavrin, D.; Zelikov, V.; Novikova, T.; Ivetić, V. The effect of low-intensive coherent seed irradiation on germinate growth of Scots pine and sugar beet. J. For. Sci. 2021, 67, 427-435, doi:10.17221/56/2021- JFS).
(Razzak et al., 2017. Differential response of Scots pine seedlings to variable intensity and ratio of red and far‐red light. Plant Cell Env. 40:1332-1340),
5. The most serious flaw of this paper is no statement available as to how seeds were collected, stored and used in terms of preventing seeds from natural sun light outdoor and fluorescent light indoor. Phytochromes respond to far-red and red instantaneously upon the exposure. Experiments the authors have done must require careful handling of seeds in a dark room (Sasaki et al. 1976, Shropshire 1973). As far as I understand from the materials and methods, sugar beet is from seed collection stored by an institute, and the author do not know how seeds were collected. If the experiment were to be carried out in more stringent ways, seeds should be surface sterilized, as well, because the irradiation could influence microbiomes associated with seeds, biological factors that should influence the behavior of seeds and seedlings, and are definitely affected by irradiation. The way they carried out the experiment does not tell us about the irradiation effect that is due to changing plant physiology, or changing microbiomes that help seeds germinate or seedlings grow, or that have detrimental effect on seed germination and growth.
6. I find figures hard to understand. For example, Fig. 2 and 3 tell us what? The images are blurred, the number of seedlings is 9, then 10, then 12. This is very confusing.
7. For the reliability of the experiment, it is necessary to measure the length and mass not selectively, but in a successively, at least 100 seedlings in each group, which were exposed to different light. Instead of the graphs shown in Figure 1, there should be boxplots, since the indicators of seedlings quite often have a median that does not coincide with the arithmetic mean and the standard deviation cannot be symmetrical.
8. Please provide a direct link to the data set for constructing Figure 1 or the data itself for verification.
9. The given germination indicators are not sufficient to assess the completeness of the study: the following must be calculated: germination percentage (GP), mean germination time (MGT), germination rate (GR), germination uniformity (GU), and healthy seedling percentage (HS) were calculated according to the following formulae
Ranal, M.A.; Santana, D.G.d. How and why to measure the germination process? Braz. J. Bot.2006, 29, 1-11.
– Germination percentage (%) = N/S×100
– Mean germination time (day) = Σ(Dx⋅nx)/N
– Germination rate (% day-1) =100 × Σ(nx/Dx)
– Germination uniformity = Σ[(Σ(Dx⋅nx)/N)-Dx)2×nx]/(N-1)
– Healthy seedling percentage (%) = Number of healthy seedlings/S×100
where D is the number of days after planting, N is number of final germinated seeds, 141 n is number of germinated seeds on day D, S is number of planted seeds.
10. At the moment, the statistical results are scattered. To achieve this goal and reliably confirm the relationship between the physiological parameters of seeds and genetic ones, it is necessary to conduct a cluster analysis and build a consensus diagram of similarities and differences for seeds that have germinated under the influence of light of different spectral composition.
11. Since the light was not coherent, how to draw a statistically reliable boundary along the wavelengths causing expression?
12. What does the term "light quality" indicated in the name of the paper mean? By what criterion is it evaluated in the paper?
Author Response
1. Abstract L16-17 – the wavelength value in nanometers is required for each region used in study.
Answer: Done.
2. It is necessary to add two sentences at the beginning of the abstract that answer the question: Why did you do it? (2 sentences, max 50 words). Briefly outline the problem you are tackling and why it is important.
Answer:
«The photoreceptors of red light (phytochromes) and blue light (cryptochromes) impact the plant growth and metabolism. However, their action has been little studied, especially in coniferous plants. Therefore,…»
3. It is necessary to add two sentences at the end of the abstract that answer the questions: "What do the results mean in practice? and What remains unresolved?"
Answer: «We suggest that the observed phenomena of red light is the most important for germination of the plantlets and may be based on earlier and enhanced expression of auxin, cytokinin, gibberellin and jasmonate signaling genes activated by corresponding photoreceptors. The obtained results may help to improve reforestation technology; however, this problem needs further study.»
4. The keywords should include the type of plant (Latin name) whose seeds were used for the study.
Answer: Done.
5. DOI must be specified in the references.
Answer: Done.
We use Zotero for reference design; hyperlinks will appear automatically if the manuscript is accepted.
6. Introduction. It is necessary to focus more on the processes occurring under the influence of light in the seeds of gymnosperms, or to indicate more clearly the transition from the model plant – Arabidopsis – to Pinus sylvestris L.
Answer: «Gymnosperms have a special system of phytochromes, since receptors of RL are represented by PHYN, PHYO, and PHYP orthologues. In addition, they have the ability to synthesize chlorophyll in the dark due to the presence of three genes for light-independent protochlorophyllide oxidoreductases L, N, and B (chlL, chlN, and chlB), which are involved in the light-independent reduction of protochlorophyllide to chlorophyllide. On the one hand, exogenous hormones may induce plant growth by acting as mediators in the processes of light signal transduction”.
7. It should be more clearly indicated why the species Pinus sylvestris L. was chosen, since the phrase "seeds are also a convenient system for estimating the amount of protein from their transcripts" can refer to any tree species.
Answer: Done.
“Given the important economic importance of Scots pine…”
8. The phrase "We assumed that the presence of RL in the light spectrum has the greatest stimulating effect on the germination and grow parameters of Scots pine seeds" requires correction, since there are diametrical studies of the effect of RL on the germination process of Scots pine seeds, and this hypothesis is not new. It should be clearer to highlight the originality of the research goal in comparison with existing studies that RL stimulates
(Qamaruddin and Tillberg, 1989. Rapid effects of red light on the isopentenyladenosine content in Scots pine seeds. Plant Physiol. 91:5-8;
Tillberg, 1992. Effect of light on abscisic acid content in photosensitive Scots pine (Pinus sylvestris L.) seed. Plant Growth Reg. 11:147-152).
or inhibits the germination process of Scots pine seeds
(Novikov, A.; Bartenev, I.; Podvigina, O.; Nechaeva, O.; Gavrin, D.; Zelikov, V.; Novikova, T.; Ivetić, V. The effect of low-intensive coherent seed irradiation on germinate growth of Scots pine and sugar beet. J. For. Sci. 2021, 67, 427-435, doi:10.17221/56/2021- JFS).
(Razzak et al., 2017. Differential response of Scots pine seedlings to variable intensity and ratio of red and far‐red light. Plant Cell Env. 40:1332-1340),
Answer:
We are grateful to the Reviewer for such a careful reading of the manuscript. We have added the mentioned articles to our references list. We also clarified the purpose of the research to emphasize the differences from previous works.
“We want to establish which spectrum of light is able to maximize the activation of Scots pine seed germination, as well as which photoreceptor genes, transcription factors, and hormonal signaling genes underlie this activation”.
9. The most serious flaw of this paper is no statement available as to how seeds were collected, stored and used in terms of preventing seeds from natural sun light outdoor and fluorescent light indoor. Phytochromes respond to far-red and red instantaneously upon the exposure. Experiments the authors have done must require careful handling of seeds in a dark room (Sasaki et al. 1976, Shropshire 1973). As far as I understand from the materials and methods, sugar beet is from seed collection stored by an institute, and the author do not know how seeds were collected. If the experiment were to be carried out in more stringent ways, seeds should be surface sterilized, as well, because the irradiation could influence microbiomes associated with seeds, biological factors that should influence the behavior of seeds and seedlings, and are definitely affected by irradiation. The way they carried out the experiment does not tell us about the irradiation effect that is due to changing plant physiology, or changing microbiomes that help seeds germinate or seedlings grow, or that have detrimental effect on seed germination and growth.
Answer: Seeds of Scots pine were provided by the Training and Experimental Forestry Enterprise of the Bryansk State Technological University of Engineering (Bryansk, Russia). The seeds were removed from the cones in a cone dryer and stored in hermetically sealed glass bottles at +4 °C in the dark. Previously, we used surface sterilization of seeds in 13% H2O2 in 96% ethanol for 5 min (Ivanov et al., Russian Journal of Plant Physiology, 2011). However, as our long-term experiments have shown, when using high-quality seeds and maintaining storage conditions, this procedure is not needed. Sowing qualities of seeds are preserved, and plantlents are not damaged by pathogenic flora. The cultivation methods used allow us to obtain pine plants of the required quality for various research purposes. See for example: Ivanov et al., Plant Physiology and Biochemistry, 2016; Ivanov et al., Environmental Science and Pollution Research, 2016; Ivanov et al., Environmental Pollution, 2018; Zlobin et al., Photosynthesis Research, 2019; Pashkovskiy et al., Plant Physiology and Biochemistry, 2019; Ivanov et al., Environmental and Experimental Botany, 2019; Ivanov et al., Environmental Science and Pollution Research, 2021; Pashkovskiy et al., Cells, 2021; Ivanov et al., Cells, 2022).
10. I find figures hard to understand. For example, Fig. 2 and 3 tell us what? The images are blurred, the number of seedlings is 9, then 10, then 12. This is very confusing.
Answer: We are grateful to the reviewer for the identified misstatement. We agree that scans of plantlets at different stages of development are redundant and illustrative. In this regard, we reduced the number of figures in the revised version of the manuscript. We also want to note that the quality of figures in the PDF file for peer review does not correspond to the submitted images. However, if the manuscript will be accepted, the figures in the maximum resolution without compression will be used.
11. For the reliability of the experiment, it is necessary to measure the length and mass not selectively, but in a successively, at least 100 seedlings in each group, which were exposed to different light. Instead of the graphs shown in Figure 1, there should be boxplots, since the indicators of seedlings quite often have a median that does not coincide with the arithmetic mean and the standard deviation cannot be symmetrical.
Answer: The number of biological replicates used in the determination of different parameters is indicated in subsection 4.6. Specifically, for Figure 1, the number of biological replicates ranged from 3 to 6. The plantlets were collected randomly, not selectively. The values presented in the tables and figures are the arithmetic means ± standard errors. We do not agree that at least 100 plantlets were needed for reliability of the experiment, since statistical analyses of data were performed with one-way ANOVA followed by Duncan's method for normally distributed data and Kruskal‒Wallis one-way ANOVA on ranks followed by the Student–Newman‒Keuls post hoc test for nonnormally distributed data and data with unequal variance.
The length and other morphometric parameters were measured only on 9-day-old plantlets, and the number of biological replicates was 20. With this number of replicates, the experimental accuracy (Px) in the Dark variant with the highest variability did not exceed 4.8%, while in other variants, the experimental accuracy did not exceed 2% (WFL, WL, BL) and 3% (RL and FRL). Therefore, increasing the number of replicates did not make sense.
12. The given germination indicators are not sufficient to assess the completeness of the study: the following must be calculated: germination percentage (GP), mean germination time (MGT), germination rate (GR), germination uniformity (GU), and healthy seedling percentage (HS) were calculated according to the following formulae
Ranal, M.A.; Santana, D.G.d. How and why to measure the germination process? Braz. J. Bot.2006, 29, 1-11.
– Germination percentage (%) = N/S×100
– Mean germination time (day) = Σ(Dx⋅nx)/N
– Germination rate (% day-1) =100 × Σ(nx/Dx)
– Germination uniformity = Σ[(Σ(Dx⋅nx)/N)-Dx)2×nx]/(N-1)
– Healthy seedling percentage (%) = Number of healthy seedlings/S×100
where D is the number of days after planting, N is number of final germinated seeds, 141 n is number of germinated seeds on day D, S is number of planted seeds.
Answer: We thank the reviewer for the valuable recommendations, which we will take into account in future work. The terms used by us for accounting for seed vigor and seed germination are calculated specifically for the seeds of Scots pine (GOST 13056.6-97. Seeds of trees and shrubs. Method for determining germination) and these two parameters are used both in forestry practice and in scientific research: Bose et al., Plant Cell and Environment, 2020; Staszak et al., PLOS One, 2020; Makhniova et al., Environmental Monitoring and Assessment, 2019. We agree with the reviewer that the data is important, but if checked for each of the light options, then this will be a separate study on the properties of pine seeds.
13. At the moment, the statistical results are scattered. To achieve this goal and reliably confirm the relationship between the physiological parameters of seeds and genetic ones, it is necessary to conduct a cluster analysis and build a consensus diagram of similarities and differences for seeds that have germinated under the influence of light of different spectral composition.
Answer: We are grateful to the reviewer for a fair comment on the manuscript. We have changed the figures in such a way that they are easier to read. In Figure 1, we changed the OX scale to time, and in the figures concerning transcripts of the genes, we changed the color scale to two colors to make the data easier to read and compare. We hope that the changes made will satisfy the reviewer.
14. Since the light was not coherent, how to draw a statistically reliable boundary along the wavelengths causing expression?
Answer: Artificial light for seedling cultivation is incoherent, and we used light that mimics the light normally used for cultivation. Coherent light is a completely different condition that requires a special approach. As a control, we can take fluorescent light, which is less coherent than white LED. A diode narrow band light (red, blue, far red) is more coherent than a diode white light. Darkness cannot be used as a control, as it is the absence of any kind of light, regardless of its coherence.
15. What does the term "light quality" indicated in the name of the paper mean? By what criterion is it evaluated in the paper?
Answer: We are grateful to the reviewer for the remark made. To avoid confusing readers, we have replaced the words “light quality” with “light of different spectral composition”.
Round 2
Reviewer 1 Report
Revision manuscript was described clearer in introduction and result sections , and the figure and legend also modified. I have further questions.
One minor comment:
Line 422, “in a refrigerator at +4 C” should be 4 ℃
Reviewer 2 Report
The authors have improved the manuscript.
Before proofreading, if such a decision is made, it is desirable in the manuscript:
1. Move the M&M section before the Results, which will be more convenient and logical for readers to understand the paper.
2. Add this phrase to the subsection “Experimental Design”, and references to the list at the end of the article. Decipher the abbreviation GOST.
«The terms used by us for accounting for seed vigor and seed germination are calculated specifically for the seeds of Scots pine (GOST 13056.6-97. Seeds of trees and shrubs. Method for determining germination) and these two parameters are used both in forestry practice and in scientific research: Bose et al., Plant Cell and Environment, 2020; Staszak et al., PLOS One, 2020; Makhniova et al., Environmental Monitoring and Assessment, 2019».